# The implementation and effectiveness of multi-tasked, paid community health workers on maternal and child health: A cluster-randomized pragmatic trial and qualitative process evaluation in Tanzania

**Colin Baynes**[1]*, **Almamy Malick Kanté**[2], **Amon Exavery**[3], **Kassimu Tani**[4], **Gloria Sikustahili**[4], **Hildegalda Mushi**[4], **Jitihada Baraka**[4], **Kate Ramsey**[5], **Kenneth Sherr**[1], **Bryan J. Weiner**[1], **James F. Phillips**[6]

1 Department of Global Health, University of Washington, Seattle, WA, United States of America,
2 Department of International Health, Johns Hopkins University, Baltimore, MD, United States of America,
3 Pact/Tanzania, Dar es Salaam, Tanzania, 4 Ifakara Health Institute, Dar es Salaam, Tanzania, 5 Scope Impact, Brooklyn, NY, United States of America, 6 Department of Population and Family Health, Columbia University, New York, NY, United States of America

* colin.baynes@gmail.com

**Data Availability Statement:** The ethical approvals from the Ethical Committee at the National Institute

## Abstract

Community health worker programs have proliferated worldwide based on evidence that they help prevent mortality, particularly among children. However, there is limited evidence from randomized studies on the processes and effectiveness of implementing community health worker programs through public health systems. This paper describes the results of a cluster-randomized pragmatic implementation trial (registration number ISRCTN96819844) and qualitative process evaluation of a community health worker program in Tanzania that was implemented from 2011–2015. Program effects on maternal, newborn and child health service utilization, childhood morbidity and sick childcare seeking were evaluated using difference-in-difference regression analysis with outcomes measured through pre- and post-intervention household surveys in intervention and comparison trial arms. A qualitative process evaluation was conducted between 2012 and 2014 and comprised of in-depth interviews and focus group discussions with community health workers, community members, facility-based health workers and staff of district health management teams. The community health worker program reduced incidence of illness and improved access to timely and appropriate curative care for children under five; however, there was no effect on facility-based maternal and newborn health service utilization. The positive outcomes occurred because of high levels of acceptability of community health workers within communities, as well as the durability of community health workers' motivation and confidence. Implementation factors that generated these effects were the engagement of communities in program startup; the training, remuneration and supervision of the community health workers from the local health system and community. The lack of program effects on maternal and newborn health service utilization at facilities were attributed to lapses in the availability of

for Medical Research (NIMR) in Tanzania and the Ifakara Health Insitute do not give us license to make the data set publicly available. However, data may be made available on request to the Ifakara Health Institute Ethics Committee contact person: Dr Mwifadhi Mrisho (mmrisho@ihi.or.tz) at the department of Health System Impact Evaluation, Plot 463, Kiko Avenue, Mikocheni, PO Box 78373. When referencing the data please use the following citation: Phillips JF, Hingora A. Introducing Community Health Agents to Accelerate Achievement of MDG 4 and 5 in Tanzania - The Connect Project. The Ifakara Health Institute, Dar es Salaam, Tanzania. Available from: https://data.ihi.or.tz/index.php/catalog/Impact-evaluation/about.

**Funding:** This study was funded through a joint funding initiative by the United States based Doris Duke Charitable Foundation through grant number DDCF2009058a (JFP) and the United Kingdom based charity Comic Relief through grant number 112259 (KT). The funders had no role in designing or implementing the program that was evaluated through this study, nor in the design of the study, data collection and analysis, decision to publish or preparation of the manuscript.

**Competing interests:** The authors have declared that no competing interests exist.

needed care at facilities. Strategies that strengthen and align communities' and health systems core capacities, and their ability to learn, adapt and integrate evidence-based interventions, are needed to maximize the health impact of community health workers.

## Introduction

The evidence that community-based primary health care (PHC) interventions can improve maternal, neonatal and child health (MNCH) has increased over the past few decades [1]. This owes, in large part, to studies that focus on specific tasks of community health workers (CHW), such as activities related to safe motherhood [2–4]; child health [5–8]; family planning [9, 10]; infectious and non-communicable diseases [11–15]; neglected tropical diseases [16] and mental health [17, 18]. In recognition of their effectiveness, there have been numerous publications collating and comparing CHW experiences across countries [19–22], reporting the costs and cost-effectiveness of CHW models [23, 24], synthesizing evidence on the design and effects of CHW programs [25–28] and developing conceptual understandings of how to scale up and integrate them into health systems [29–35]. With this, opinions converged in favor of deploying CHW, with increasing numbers of proponents citing evidence that access to essential, low-cost interventions from these cadres can help end preventable mortality, particularly among children in low- and middle-income countries (LMIC) [36].

Accordingly, in recent decades there has been a rapid expansion of CHW deployment in many countries. Despite this, respected observers have argued that the use of CHW remains an underdeveloped component of health systems in LMIC [1]. Indeed, studies on the implementation of CHW programs have noted that support for CHW, their performance and integration into communities and health systems is uneven across and within countries [37, 38]. Frequently, evidence-based recommendations are not effectively applied as CHW programs are designed and implemented [30, 35, 39], and many CHW programs are fraught with challenges including poor planning; unclear or fragmented roles; inadequate training; weak supervision; lapses in logistical processes; tenuous accountability linkages; ineffective incentive structures; poor selection processes and dissatisfaction of communities [40–43].

The impact of CHW can be maximized if implementers and policymakers understand the reasons for these problems well and adapt implementation strategies for deploying CHW accordingly. However, most research has focused on the effects of CHW on service utilization and population health with less attention to implementation process [44], and the contextual factors that influence success [45, 46]. Studies of CHW program implementation are often detached from rigorous evaluations of programs' impact [47]. To date there is limited evidence on CHW effectiveness from randomized studies delivered through routine health systems [48, 49], and, furthermore, there is a lack of pragmatic evidence on CHW programs that achieved mixed results even though analyses of barriers to implementation success are needed to help programs [25]. In this paper, we share results from a cluster-randomized pragmatic implementation trial of a CHW program in Tanzania that was conducted from 2011–2015. This trial, called *Connect*, evaluated the impact of a CHW program on child survival, the primary outcome; and on MNCH behaviors, childhood morbidity and care seeking for sick children, the secondary outcomes. Since previous publications on *Connect* have reported that WAJA implementation had no statistically significant effect on childhood mortality [50], we report here the impact of the intervention on secondary outcomes and findings from an embedded qualitative process evaluation conducted from 2012–2014.

## Materials and methods

### Ethical considerations

Approval for the *Connect* trial was granted by the ethical review boards of the IHI (IHI/IRB/ No. 16–2010), the National Institute for Medical Research of Tanzania (NIMR/HQ/R.8a/Vol. IX/1203), and the Institutional Review Board of Columbia University Medical Center (Protocol AAF3452).

### Study setting

*Connect* was situated in the sentinel areas of the Ifakara and Rufiji Health and Demographic Surveillance Systems (HDSS) managed by the Ifakara Health Institute (IHI). The Ifakara HDSS is in Morogoro, a landlocked region in southwestern Tanzania, and traverses two districts, Kilombero and Ulanga, that are approximately 500 km by road from Dar es Salaam, Tanzania's largest city. The Rufiji HDSS is in Rufiji district on the Indian Ocean coast approximately 150 km south of Dar es Salaam by road. The population under surveillance in 2015 was approximately 380,000 (280,073 in Ifakara and 99,206 in Rufiji) [51, 52]. The communities under surveillance in Ifakara and Rufiji are predominantly rural and rely on subsistence farming, however within the sentinel areas in each district there are small peri-urban areas with businesspeople and traders.

### Intervention background and design

The use of CHW has precedent in Tanzania [53]. Since their introduction in the 1970s, the deployment of volunteer CHW proved fraught with implementation and maintenance problems and failed to provide evidence that unpaid workers could provide effective and sustainable means to extend PHC to communities [54].

In 2007, the Government of Tanzania promulgated the Primary Health Care Services Development Plan, known in Swahili as *Mpango wa Maendeleo wa Afya ya Msingi* (MMAM), which called for the revitalization community PHC by way of establishing a national cadre of paid, multi-tasked CHW [55]. In 2010, UNICEF and the Ministry of Health and Social Welfare of Tanzania (MOHSW) carried out a situation analysis of existing CHW programs in the country to inform recommendations on strategy for operationalizing the MMAM vision. Their report recommended that the national CHW, which they named *Wawezashaji wa Afya ya Jamii* ("community health enablers" or "WAJA"), be selected by their communities, formally trained and enrolled in national health sector scheme of service, accorded a salary and government recognition and tasked with performing an integrated package of MNCH services [56]. Yet, at this time, there was no experience of operationalizing such a program, nor systematic evidence on whether the proposed CHW model would be acceptable, feasible and cost-effective or have an incremental impact on MNCH. In 2010, the IHI, MOHSW, the Tanzania Training Center for International Health and Mailman School for Public Health at Columbia University launched *Connect* to address that knowledge gap.

**CHW recruitment, selection and training.** The design of the *Connect* intervention is described in depth elsewhere [57, 58]. In 2010, IHI staff and Council Health Management Teams (CHMT) in Kilombero, Ulanga and Rufiji oriented community leaders to the intervention and recruited community members to stand for election to become WAJA. Community members could become a WAJA if they had received form four level education attainment (US grade 10) and passing grades in science, as per requirements for government employment, and longstanding residency in their current home village. Candidates whose eligibility was

confirmed by local authorities were given two-three weeks to make their case to the community as candidates, and after that villages held elections to select their WAJA.

In October 2010, candidates chosen by their communities went to Ifakara to undertake training of nine months, the minimum duration required for government employment. The curricula received national accreditation and comprised of two semesters of didactic and practical, clinic-based training in human biology; basic clinical skills; health promotion and disease prevention in the community; sexual and reproductive health; integrated management of childhood illness (IMCI); management of basic pharmaceuticals; stakeholder mapping and networking and community mobilization; and a community-based practicum. The WAJA service package was developed as a strategy to expand access to education on MNCH, mobilize villages to collectively promote community health, enhance referrals and utilization of antenatal care (ANC), facility-delivery by skilled birth attendants, postnatal care (PNC), and make IMCI available in communities. WAJA also distributed oral contraceptives and condoms and performed basic curative care for gut infections caused by worms, fever, and non-complicated cases respiratory infection, diarrhea, and malaria. See Fig 1 [59].

**CHW deployment and implementation management.** WAJA received employment contracts from local government authorities upon their graduation from training. This accorded them a salary of approximately $112 (United States dollar) per month. *Connect* developed a two-tiered system for supervising WAJA to promote the clinical quality of care, and to ensure that WAJA were accountable to communities. During training the CHMT appointed nurses or clinical officers to be 'facility supervisors' of WAJA that were deployed to communities in their facilities' catchment areas. Village Health Committees from intervention communities appointed a 'village supervisor'. Both supervisors participated in practical

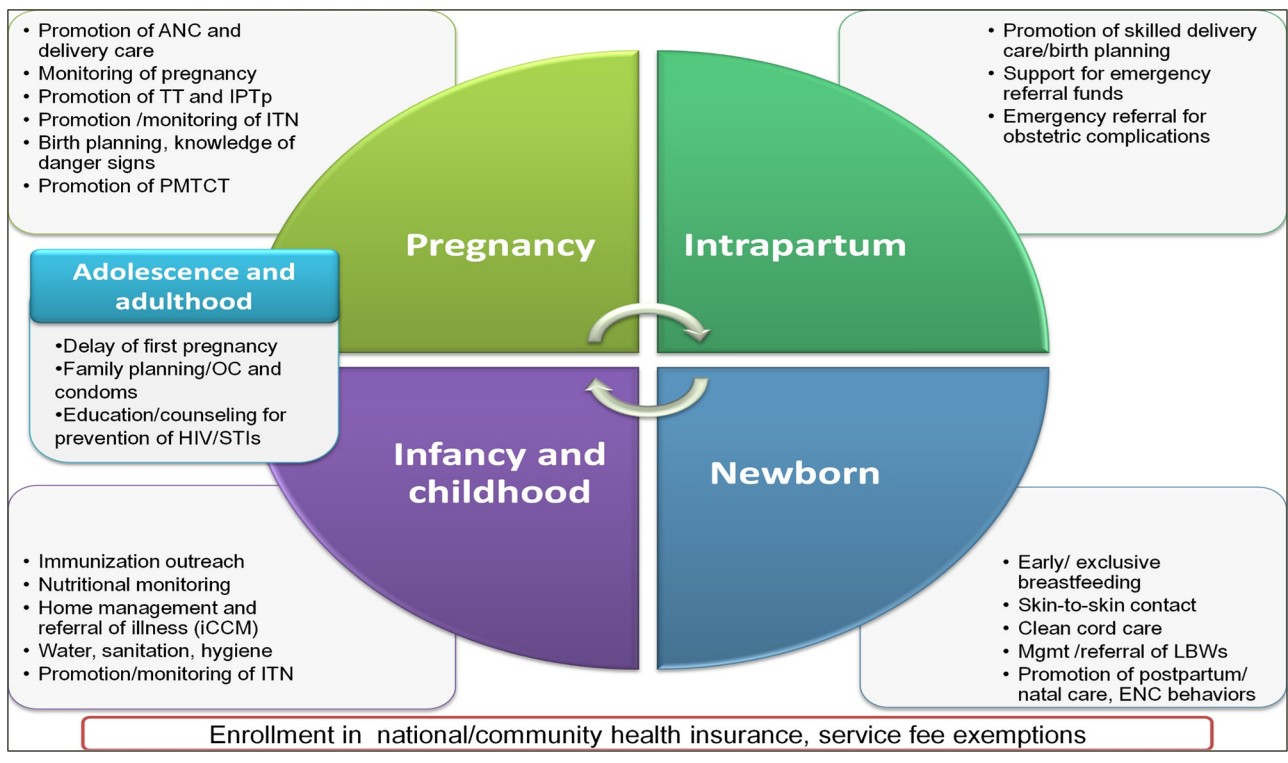

**Fig 1. The *Connect* WAJA service delivery package.**

sessions of the WAJA training and participated in workshops with IHI staff on the WAJA role and work package, supervision and mentoring and community engagement.

Implementation arrangements of *Connect* balanced the overriding goal of understanding the effects of WAJA deployment through the routine health system with the reality that to deliver the program, the health systems required support from development partners [60]. To provide this support, the IHI issued a financial subaward to Tanzania Training Center for International Health to develop and implement the training program. The IHI also issued subawards to the respective CHMTs to cover expenses associated with WAJA recruitment and selection; field practicum; the orientation of WAJA supervisors and their performance of timely supervision; health information system staffs' compilation of routine service delivery data recorded by WAJA; procurement and distribution of WAJA equipment, supplies, and medicines; and payment of WAJA salaries. In addition, the IHI seconded Implementation Coordinators to the CHMT to help manage the deployment of WAJA and the launch of their activities in communities and establish processes for routinizing the provision of management and health system supports to the new cadre.

The provision of this assistance lasted for the first two years of the trial (2011–12) after which, the IHI withdrew direct support for implementation. At this point, CHMT were to obtain resources for maintaining the WAJA in their posts through the routine comprehensive council health planning process. From 2013 onward, the IHI led the evaluation of the WAJA intervention only, setting aside resources to ensure availability of remuneration, supervision, and essential supplies to WAJA if CHMTs failed to secure resources through the health system. *Connect* never intervened to influence the readiness or ability of facilities to provide referral level care. WAJA, 142 in total, were recruited, selected, and trained in two cohorts, the first deployed to 25 intervention villages in August 2011 and the second to the remaining 25 intervention villages in August of 2012.

## Outcome evaluation

**Study design.** The detailed protocol for the *Connect* trial (registration number ISRCTN96819844) has been published elsewhere, including CONSORT checklist for pragmatic trials (S1 Table) [57]. *Connect* was a cluster-randomized pragmatic trial in the 101 villages within the areas of the Ifakara and Rufiji HDSS (63 in Ifakara, 38 in Rufiji). Stratified randomization was used to allocate 50 villages to the intervention arm and 51 to the comparison arm. The unit of randomization was the village. In 2010, a public drawing was organized to randomly assign villages to the two arms. Villages were block-randomized within the strata defined by village population size. Stratification was segmented by four categories to achieve a 1:1 match of communities in each arm that had <1000 population, 1000–2999, 3000–4999, and ≥5000. Local government and village leaders attended the drawing, selecting representatives to draw pieces of paper with the name of each village written on them from containers numbered for each stratum. The villages chosen by the representatives were to be randomized to the intervention and comparison arms, and which representative represented which arm and the villages that they picked were concealed until after the drawing. Within the intervention arm, villages received between one and four WAJA depending on their strata. Villages allocated to the comparison received the 'standard of care' which comprised of routine activities coordinated by village governments to promote community health and households' recourse to facility-based care for preventive or curative care. Because of the nature of the intervention, it was not possible to mask participants to their treatment status.

**Data sources and sampling.** The HDSS provided the platform to ascertain *Connect's* primary outcome, childhood mortality. As this has been published elsewhere, we report here the

effect of WAJA on secondary outcomes, MNCH behaviors and incidence of childhood disease. To obtain data for the evaluation of secondary outcomes, the IHI conducted household surveys in villages in the 101 villages before deployment of WAJA in April-August 2011 and after four years of implementation, April-August 2015.

HDSS censuses conducted in 2010 and 2014 provided the sampling frame for the pre- and post-intervention surveys. Calculations of data from a survey conducted in the same districts the year before indicated that to detect a minimum of 12% difference in the prevalence of secondary outcomes, the surveys needed to enroll per village a minimum of eight women that had had a live birth in the previous two years, and that to detect a minimum of 5% difference between arms in the incidence of childhood diarrheal or febrile/respiratory illness, the surveys would have enroll per community primary caregivers of at least 15 under-five year-old children (assuming $\alpha = 0.05$, $\beta = 0.80$, $k = 0.25$ and two-tailed test).

*Connect* researchers employed 'probability proportional to size' techniques and used census data to randomly select households for recruiting survey participants [61]. Participants were eligible if they were a female between the ages of 18–49 or the primary caregiver of an under-five year-old child and resident in the household that was randomly selected. In the end, both baseline and endline household surveys met the minimum sampling requirements, enrolling 3,267 and 3,048 women aged 18–49, respectively, including 882 and 778 mothers that had delivered a live birth in the previous two years. The participants reported on 2,104 under-five year-old children at baseline and 1,565 at endline. Over time, no adverse events were reported in either trial arm. See Figs 2 and 3, the participant flow diagrams.

**Data collection and management.** The data collection team comprised of Tanzanian research assistants with degrees in public health, sociology, or other relevant disciplines. They underwent a one-week training, which was followed by two days of pre-testing during which

**Baseline Household Survey (2011)**

**Fig 2. Baseline participant flow diagram.**

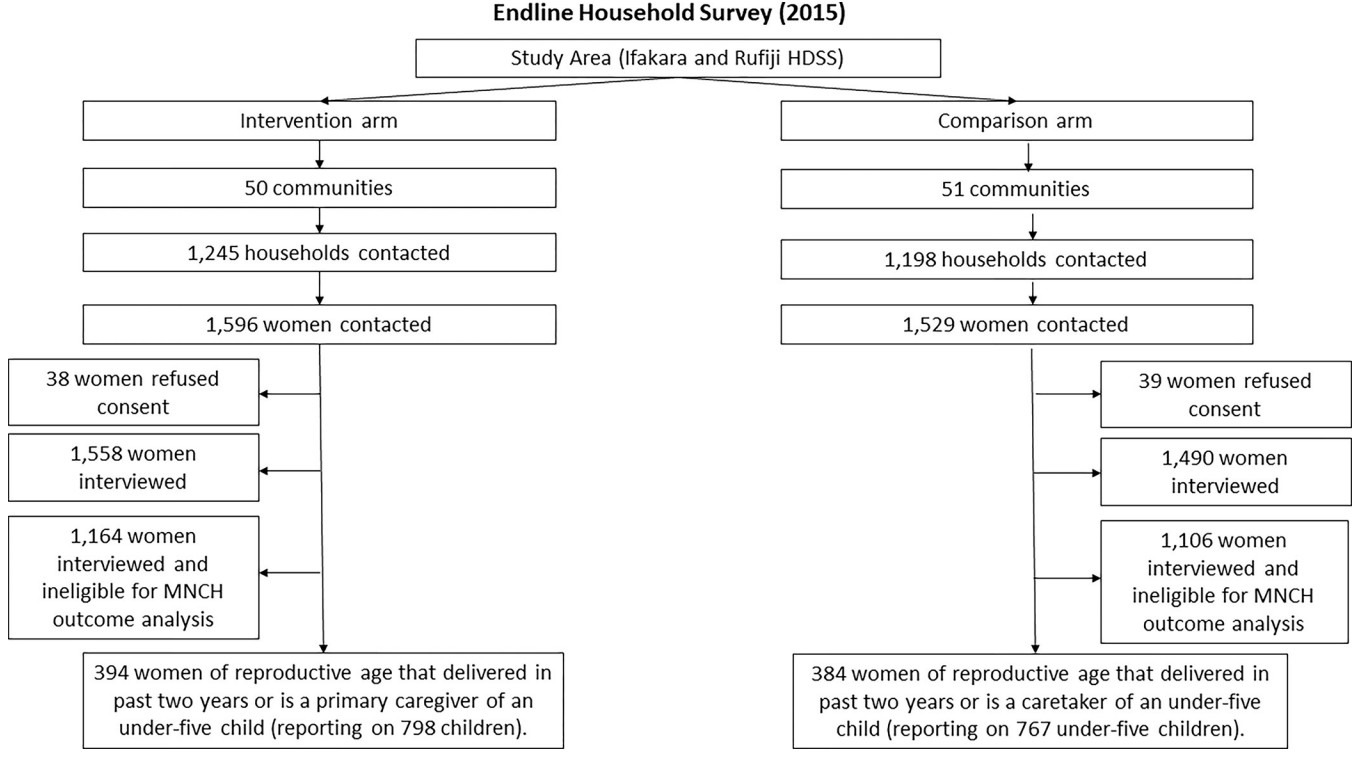

**Fig 3. Endline participant flow diagram.**

all data collectors administered the full survey at least twice. *Connect* staff divided the data collection team into groups and assigned each group to clusters of communities. Data collectors recruited all women of reproductive age that were resident in each of the selected households no matter their childbearing history. Those that agreed to participate were read aloud an informed consent form (ICF). Individuals that agreed to participate either signed the ICF or provided an inked thumbprint to confirm their agreement to participate. Data collection was paper based. *Connect* embedded staff members in each data collection group to review completed surveys to assure their quality and pass them on for data entry into an Epi-Info database where they were cleaned and prepared for analysis. Information that could identify individual participants during and after data collection was available on the ICF which contained participant names and unique identifiers. ICF were stored in a secure, locked environment at the IHI office and not accessed by the authors during data analysis.

**Outcome measures.** This study reports on the effect of WAJA deployment on secondary outcomes listed in Table 1. All outcomes were self-reported by household survey participants and were specified before the trial. Outcomes one to five in Table 1 refer to service utilization and health behaviors practiced by mothers with respect to their most recently born child if the child was born within two years prior to the survey. Outcomes six and seven refer to the incidence of childhood morbidities in the two weeks prior to the survey among all children under five years of age whose mothers or other primary care givers participated in the survey. Outcomes eight to 10 refer to the timeliness and appropriateness of care received by all under-five children if their mother or caregiver reported that they had become ill with diarrheal, febrile, or respiratory sickness during the two weeks before the survey.

**Data analysis.** For each outcome, we estimated the effect of the intervention using an intent-to-treat approach and logistic difference-in-difference (DiD) regression analysis with

**Table 1. Secondary outcomes measured through *Connect* household surveys (2011 and 2015).**

| | |
|---|---|
| 1 | First trimester antenatal care (ANC) initiation |
| 2 | 4+ ANC sessions |
| 3 | Facility delivery |
| 4 | Exclusive breastfeeding |
| 5 | Postnatal care (newborn) |
| 6 | U5 diarrhea incidence |
| 7 | U5 cough/fever/difficulty breathing incidence |
| 8 | Oral rehydration therapy (ORT) for children with diarrhea |
| 9 | Malaria test for febrile children |
| 10 | Appropriate care (Artemisinin Combined Therapy (ACT) for malaria or antibiotics for respiratory illness.) for children with febrile or respiratory symptoms |

fixed effects for time and trial arm and random-intercepts to account for clustering of observations within villages. See Eq 1:

$$logit(y_{ijk}) = \beta_0 + \beta_1 time_k + \beta_2 int_{jk} + \beta_3(time_k \times int_k) + u_{ij} \tag{1}$$

where $y_{ijk}$ is the log of the odds that the *kth* individual in the *jth* cluster in the *ith* treatment arm experienced the outcome (Table 1), *time* is an indicator of data source of observation $k$ whereby *time* equals zero for baseline and one for endline, *int* is the intervention group dummy for individual $k$ in cluster $j$, and *time × int* is the DiD indicator of the time by intervention interaction. $\beta_3$ is the DiD estimator for the effect of WAJA deployment on outcomes above and beyond changes associated with the passage of time.

Given the balance of the sample before the intervention (Supplemental File 2), we did not adjust for individual-level characteristics of participants. In the model, $u_{ij}$ is the random effect corresponding to the *jth* cluster in the *ith* trial arm and is normally distributed. Because the combination of binomial variation within clusters and normal variation between clusters, quadrature methods were used to maximize the likelihood and obtain parameter estimates, cluster robust standard errors and confidence intervals and conduct the significance test for this model. The DiD approach relies on the 'parallel trends' assumption. While we could not formally test this assumption, we used data from the HDSS to verify that child mortality trends in the two arms were similar during the 10 years prior to the trial.

## Process evaluation

**Qualitative study design.** In July and August 2012, 2013, and 2014, respectively, we carried out a qualitative process evaluation. The goal of this was to understand the changes that arose in the local health system and communities as a result introducing WAJA, whether those changes led to proximal outcomes associated with desired health and behavioral changes, and how contextual factors shaped that process. We situated the qualitative process evaluation within the same 'nodes' in each year, which we defined as intervention villages, their aligned PHC facilities which received WAJA referrals and provided WAJA supervision, and their respective CHMT.

We purposively sampled four nodes in Ifakara and two in Rufiji. We sampled one urban and three rural nodes in Ifakara and one urban and one rural node Rufiji. Participants from the community-level were parents of under-five year-old children, WAJA, village supervisors of WAJA, Village Executive Officers and Village Chairpersons. Participants from the health system included WAJA health facility supervisors, District Medical Officers, District

**Table 2. Qualitative data collection methods and participants.**

| Informant Type | Year and data collection method | | | | | |
| --- | --- | --- | --- | --- | --- | --- |
| | 2012 | | 2013 | | 2014 | |
| | IDI | FGD | IDI | FGD | IDI | FGD |
| WAJA | 5 | 2 | 4 | 4 | 8 | 0 |
| Other community stakeholder | 3 | 4 | 5 | 0 | 6 | 0 |
| Health facility staff | 4 | 2 | 4 | 2 | 5 | 0 |
| District-level health management staff | 5 | 0 | 6 | 0 | 6 | 0 |

Reproductive and Child Health Coordinators and District *Connect* focal persons. We analyzed transcriptions of 75 focus group discussions (FGD) and in-depth interviews (IDI). See Table 2.

**Qualitative data collection and management.** Qualitative data collectors comprised of a team of experienced Tanzanian researchers with degrees in public health, sociology or another relevant discipline who received a three day training. After this, they pre-tested the instruments in an intervention setting that were not situated in the process evaluation nodes. These steps were conducted in all three years of the process evaluation. Each IDI and FGD was conducted Swahili and facilitated by two data collectors, one that led the interview and the other that took notes. Prior to the onset of the interviews, the data collectors administered an informed consent process in which they read aloud an ICF. Participants that agreed to participate either signed the ICF or provided an inked thumbprint. All interviews and discussions were recorded on a digital device. Data collection pairs transcribed all interviews and discussions, in Swahili, within one day of completing them. Swahili language transcripts were reviewed by a qualitative specialist from the *Connect* team to assure their quality. Transcripts and ICF were maintained password encrypted electronic files or in locked cabinets at the IHI. Transcripts were then cleaned and translated into English.

**Qualitative data analysis.** To analyze the data, we first reviewed the transcripts, memoing extensively on patterns in the data, their meanings and ways in which these could be studied in a more structured analytical process [62, 63]. Based on this, we constructed causal pathway models (CPM) to develop a theory of how WAJA worked to produce the evaluation outcomes and how contextual conditions influenced that process [64]. In doing so, we identified the relationships between the following constructs in the data: elements of the *Connect* implementation strategy, proximal outcomes, the mechanisms that the strategy triggered to affect proximal outcomes, and the determinants that either helped or hindered implementation. We established these constructs as analytic themes and created codes aligned to each theme.

We uploaded 75 transcripts into *Dedoose* analytic software and coded. For this we adapted steps associated with grounded theory [65, 66]. First, we conducted 'open coding' in which we utilized codes from the 'strategy', 'outcome' and 'determinant' themes. Then, we sorted the coded transcripts by implementation outcome codes and examined relationships within and across segments of code to examine which factors exerted the most influence over implementation, whether these influences were positive or negative, and how they shaped outcomes. Using segments of text that were assigned 'open codes', we, then, pursued 'axial coding', utilizing codes from the 'mechanism' theme only. In this analysis, we defined 'mechanisms' as events or processes through which strategies produce outcomes [67]. After identifying mechanisms, we observed thematic linkages in the data with an emphasis on understanding how determinants affected the activation of mechanisms and whether they generated expected outcomes. We used coded segments of text to map findings against CPM configurations that we had developed earlier to refine them and reach conclusions about the mechanisms and determinants of WAJA implementation [68]. Finally, we triangulated our qualitative and

quantitative findings, and linked conclusions on the generative process of implementation outcomes with evidence on WAJA effectiveness. Based on this, we formulated hypotheses on the relationship between implementation dynamics and MNCH outcomes that were measured during the trial.

In the following section, we first present the quantitative analysis of WAJA effectiveness. Then we share the findings of our qualitative analysis, which is organized around four CPM that help explain how the interplay between *Connect* implementation processes and contextual factors triggered complex changes that, in turn, led to the different outcomes that we captured through survey research.

## Results

### Outcome evaluation

The characteristics of women and children across intervention and comparison arms before the trial were generally similar (S1 Table). Table 3 shows the effect of WAJA on MNCH outcomes. The intervention had no effect on the mother-level outcomes that had been established *a priori*.

Our findings indicate that the WAJA intervention improved child health. Above and beyond changes in the incidence of childhood illness that occurred with the passage of time, the reduction in the odds of diarrhea incidence was greater among under-five year-olds in intervention villages than in comparison villages ($\beta_3$ = 0.51, 95% CI: 0.32–0.81, p = 0.004). We found a similar outcome regarding the incidence of childhood febrile and respiratory illness ($\beta_3$ = 0.59, 95% CI: 0.38–0.94, p = 0.028). Findings indicate significant effects of WAJA on access to essential care and treatment for diarrheal, febrile, and respiratory sickness in children. After accounting for changes that occurred with the passage of time, we found that children in intervention villages that had diarrhea in the two weeks prior to data collection were 1.71 times more likely to receive ORT than such children in comparison settings; however, this finding was not significant at a 0.05 level ($\beta_3$ = 1.71, 95% CI: 1.06–3.18, p = 0.074). However, the effects of WAJA exposure on children with febrile and respiratory illness were greater. Compared to children with fever in comparison areas, those in intervention communities were 1.80 times more likely to receive a malaria test ($\beta_3$ = 1.80, 95% CI: 0.99–3.18, p = 0.050), and compared to children with febrile and/or respiratory symptoms in comparison communities, those in intervention communities were 1.68 times more likely to receive either antibiotic or ACT treatment ($\beta_3$ = 1.68, 1.00–2.91, p = 0.048).

### Process evaluation

Figs 4–7 are the refined CPM that illustrate the findings of the qualitative process evaluation. *Connect* deployed an implementation strategy that included nine salient components (Cells 2, Figs 4–7) to address specific modifiable factors (Cells 1) by activating mechanisms (Cells 3), which *Connect* hypothesized would generate proximal outcomes (Cells 4) on the pathway to the MNCH outcomes (Cells 5). Whether the implementation strategy components could succeed was determined by preconditions in the environment, the presence or absence of which were essential for or prevented implementation success (Cells 8 and 9). Implementation effectiveness was also moderated by contextual factors (Cells 6 and 7), which amplified or diminished the force with which mechanisms incurred the intended effect.

The qualitative analysis reveals how the *Connect* causal pathway played out at four levels [69]. The implementation strategy engaged community members in recruiting, selecting, and deploying WAJA to generate the perception that the intervention originated locally and had been adapted to meet local needs, and, with this, promote communities' acceptance of WAJA

**Table 3. Effect of WAJA deployment and implementation on MNCH outcomes.**

| Outcome | | Baseline (2011) | | | Endline (2015) | | | Difference in Difference (Impact) | | |
|---|---|---|---|---|---|---|---|---|---|---|
| | | Intervention (%, (n)) | Comparison (%, (n)) | Diff. % | Intervention (%, (n)) | Comparison (%, (n)) | Diff. % | $\beta_3$ (DiD estimator) | 95% CI | p value |
| _Mother-level outcomes[λ]_ | | | | | | | | | | |
| 1 | First trimester ANC initiation | 16 (72) | 17 (72) | 1 | 17 (67) | 24 (92) | 7** | 0.65 | 0.38, 1.13 | 0.13 |
| 2 | 4+ ANC sessions (facility only) | 41 (188) | 44 (189) | 3 | 43 (169) | 46 (177) | 3 | 0.81 | 0.39, 2.32 | 0.35 |
| 3 | Facility delivery | 71 (322) | 74 (320) | 3 | 83 (326) | 86 (331) | 3 | 0.81 | 0.46, 1.44 | 0.48 |
| 4 | Exclusive breastfeeding | 41 (186) | 39 (168) | 2 | 45 (177) | 39 (150) | 6 | 1.33 | 0.85, 2.06 | 0.21 |
| 5 | ≥1 PNC session (Facility only) | 36 (166) | 34 (147) | 2* | 25 (98) | 33 (127) | 8* | 0.83 | 0.52, 1.33 | 0.44 |
| _Child-level outcomes[σ]_ | | | | | | | | | | |
| 6 | Diarrhea in past 2 weeks | 13 (135) | 11 (124) | 2 | 6 (48) | 12 (92) | 6*** | 0.51 | 0.32, 0.81 | <0.01 |
| 7 | Febrile & respiratory symptoms in past 2 weeks | 14 (138) | 12 (127) | 2 | 5 (40) | 11 (85) | 6* | 0.59 | 0.38, 0.94 | 0.03 |
| 8 | ORT for children with diarrhea[t] | 56 (75) | 61 (75) | 5 | 54 (26) | 41 (38) | 13** | 1.71 | 1.06, 3.18 | 0.07 |
| 9 | Malaria test for febrile children[†] | 49 (171) | 55 (200) | 6* | 69 (84) | 60 (93) | 9** | 1.80 | 0.99, 3.27 | 0.05 |
| 10 | Appropriate care for children w/ febrile or respiratory symptoms[†] | 50 (182) | 49 (195) | 1 | 58 (110) | 49 (112) | 9** | 1.68 | 1.00, 2.91 | 0.05 |

***denotes statistical significance at 1%, ** 5%, * 10%

[λ]For mother level outcomes, n = 454 and 394 in intervention arm at baseline and endline; and 428 and 384 in comparison arm at baseline and endline.

[σ]For child-level outcomes, n = 1,038 and 798 in intervention arm at baseline and endline; and 1,066 and 767 in comparison arm at baseline and endline.

[t]Out of 135 and 48 children with diarrhea in intervention arm at baseline and endline; out of 124 and 92 children with diarrhea in comparison arm at baseline and endline.

[†]Out of 349 and 122 children with febrile symptoms in the intervention arm at baseline and endline; out of 396 and 155 children with febrile symptoms in the comparison arm at baseline and endline.

[†]Out of 367 and 190 children with febrile or respiratory symptoms in the intervention arm at baseline and endline; out of the 396 and 229 children with febrile or respiratory symptoms in the comparison arm at baseline and endline.

and adoption of desired behavior changes (Fig 4). _Connect_ fostered collaboration between health system and community stakeholders, designed a WAJA work package and eligibility requirements that were responsive to stakeholders' needs and consistent with health system processes. The objective of this was to align communities' and health systems' receptivity to the program, ensure the intervention was compatible with the meanings that stakeholders attached to it, and make it feasible for them to adapt to changes instigated by WAJA implementation (Fig 5). _Connect_ also built WAJAs' capacities through training, hiring and remunerating the cadre, and created systems to supervise WAJA and meet their logistical requirements. _Connect_ believed this would motivate and engender confidence in the cadre, establish PHC delivery readiness in communities, and, thereby, help achieve better MNCH outcomes (Fig 6). Finally, by providing financial and technical support to CHMT, _Connect_ sought to enable the health systems' adoption of the WAJA program as a core and sustainable component of the larger PHC system (Fig 7). These causal pathways came to fruition with mixed results.

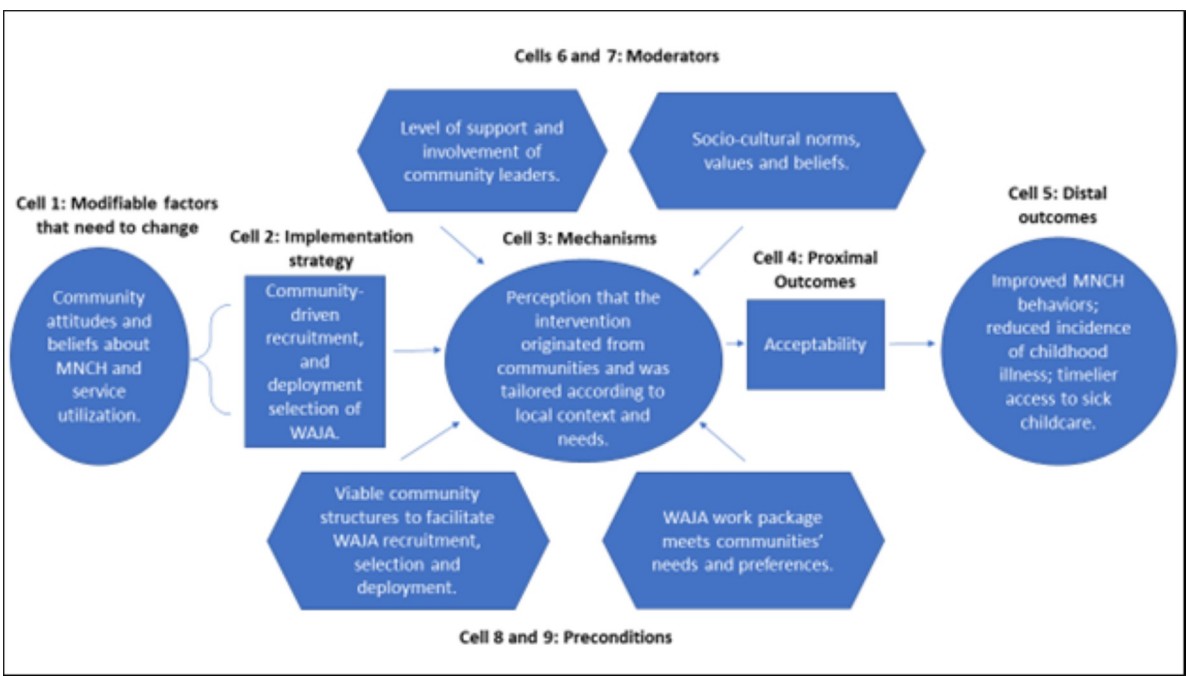

**Fig 4. Beneficiary-level causal pathway model.**

**Beneficiary level.** At the beneficiary level consistent patterns emerged across examples of community members that developed positive connections with WAJA. Community members generally referred to their relationship with WAJA in terms of a kinship bond, calling them '*WAJA, watoto wetu*' (WAJA, our children), '*vijana vyetu*' (our youths), '*wanangu*' (my child).

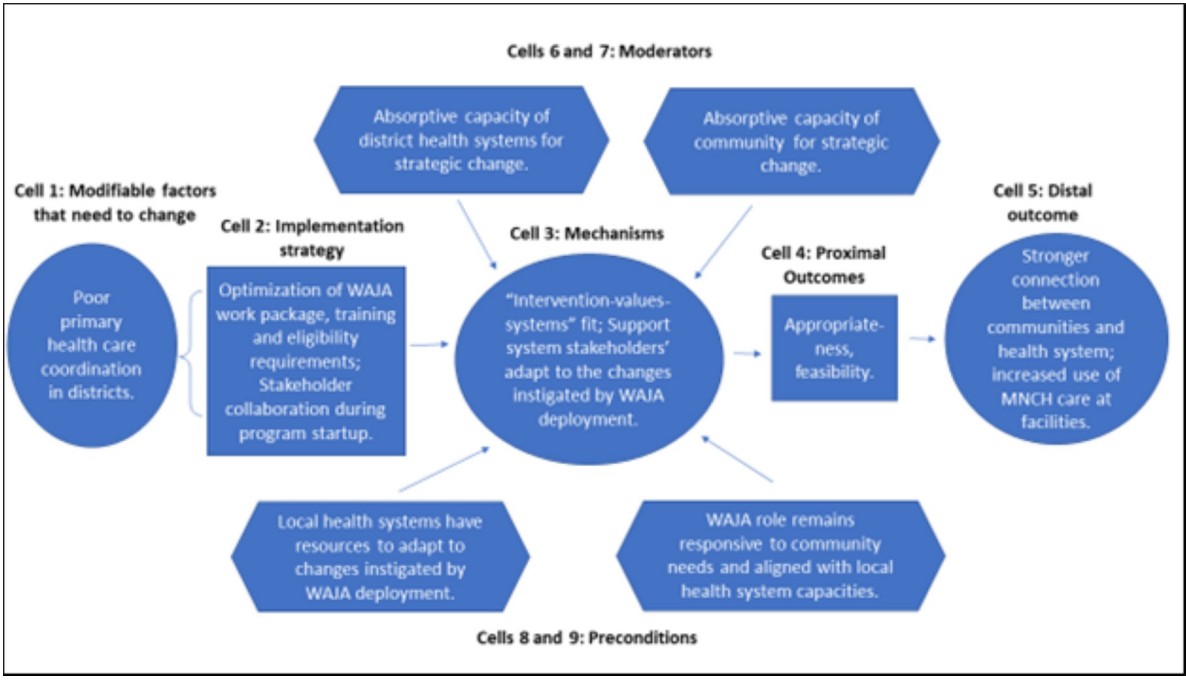

**Fig 5. Stakeholder-level (local health system and community) causal pathway model.**

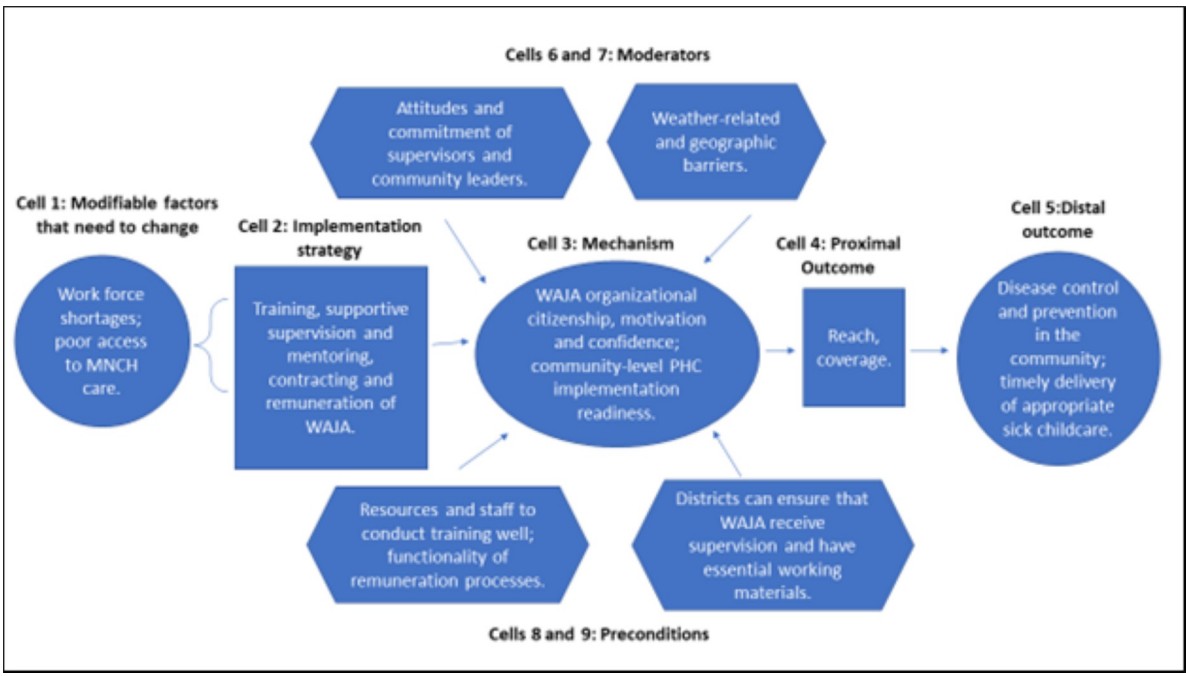

**Fig 6. WAJA-level causal pathway model.**

In addition, the participatory recruitment and selection instilled in villagers' confidence in the cadre. According to a mother: "Because of our faith in [WAJA], we selected them. So, they cannot do anything to betray us." (Mother, Lukolongo, Kilombero, 2012).

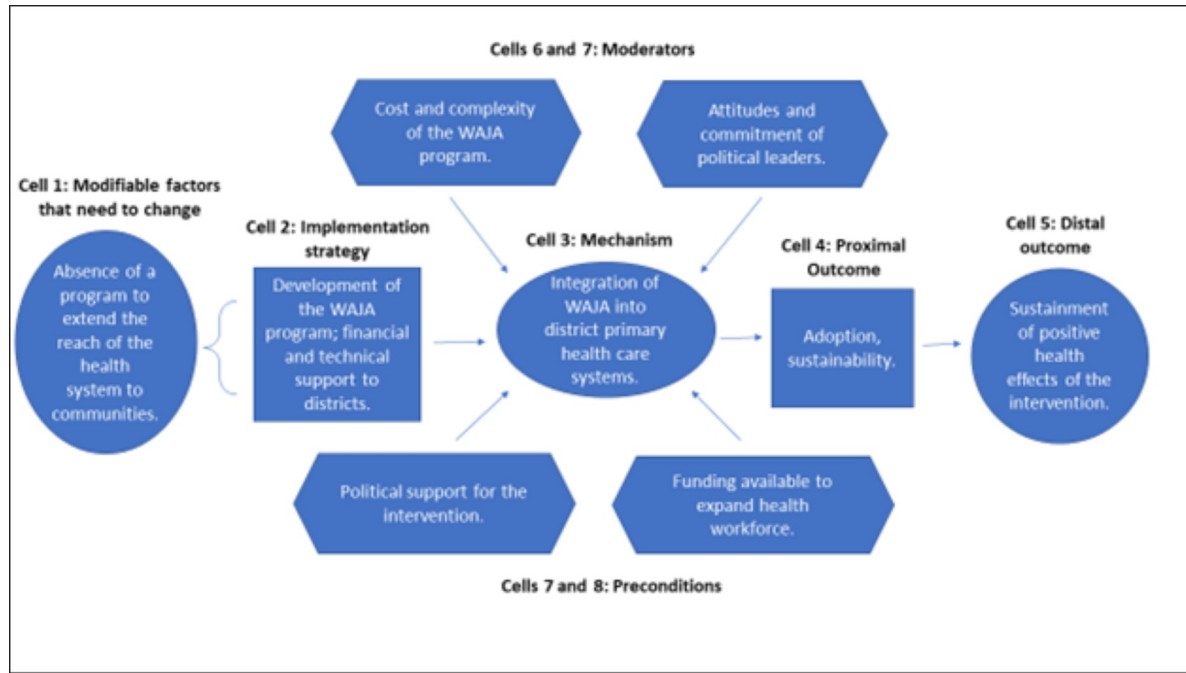

**Fig 7. Organizational-level causal pathway model.**

However, their connection to communities occasionally caused difficulty.

*One day I was educating my old friends about family planning, and they asked me 'how many are we in our family', because they know we are many. They told me 'How come you are many and now you are telling us about family planning?' That also is a problem.* (WAJA, Nyambunda, Rufiji, 2013).

Others believed that it was inappropriate for youth to get involved in reproductive health issues. A Village Supervisor reported:

*Many pregnant women are not free to show their condition. If you give them education, they says 'this [WAJA]with no family, how does he know to tell me about [being pregnant]?' (Village Supervisor, Lumemo, Kilombero, 2012).*

The qualitative data, however, illustrates that most resistance to the intervention dissipated as communities acquired experience with WAJA. This was facilitated by involvement of village leaders in introducing WAJA to communities and establishing clarity on their roles. Community members recall leaders "calling a village meeting where [leaders] say to citizens 'those WAJA who studied are back with this responsibility' and sending them to the streets where we recognize them" (Mother, Minepa, Ulanga, 2012). In other instances, WAJA reported difficulty in gaining communities' trust: "At the beginning, the biggest thing was the lack of being known [because] there was no meeting. In our village there are conflicts and there have not been village meetings in months" (WAJA, Lukolongo, Kilombero, 2012).

Over time, word spread in communities that helped establish widespread acceptability of the program. According to one early skeptic,

*"We first saw these [WAJA] and said how is it that they have become our doctors? Then we hear from more [people] that [WAJA's] medicines were good, and they give good lessons. Before when our child got sick there was rushing [to facilities], but now we do not rush to facilities anymore."* (Father, Kisawasawa, Kilombero, 2014).

Community members believed that WAJA understood them especially and tailored services to meet their needs. According to a mother:

*The method [WAJA] use is good language. If your child needs a treatment, they sit with [the caregiver] and they advise her. By telling her to go to the hospital based on the good language he used, she understands and will go rather than short answer language we don't understand."* (Mother, Mgomba North, Rufiji, 2013).

The acceptability of the WAJA created conditions in which beneficiaries could learn and adopt desired MNCH practices. Nevertheless, participants, especially men, lamented that WAJA could not perform more services.

*WAJA don't have an idea on how to treat [men]. So, I would like to see WAJA being trained more so that they can help us rather than going to the health facilities* (Father, Mgomba North, Rufiji, 2014).

As Fig 4 shows, the components of the implementation strategy that targeted factors at the beneficiary level achieved success. Preconditions for effectiveness were in place at the outset of

implementation, and *Connect* activities withstood adverse moderating factors or helped to mitigate their negative influences. Overall, the analysis suggests that change mechanisms were activated and generated the intended responses. See Fig 4.

**Stakeholder level (local health system and community).**  At the support system stakeholder level, *Connect* sought to establish an 'intervention-values-systems fit' in which members of communities and the local health system had a shared receptivity to the intervention and opportunities and abilities to adapt to the changes it introduced. This was based on the belief that if there was alignment between characteristics of the intervention, the meanings and values attached to the WAJA program by those affected by it in communities and local health system, and their absorptive capacities for strategic change, then effectiveness of the program would be greater [70, 71]. To establish a shared receptivity to the intervention, *Connect* facilitated collaboration between district employees and community members during intervention startup. According to a member of the Kilombero CHMT that served as focal person to *Connect*:

> *The village has a process to make sure that applicants belong to that village and know how to serve [the village]. Then, still the district team checks the [authenticity of] education certificates and interviews [applicants]. Then if [WAJA] are chosen by the village and complete studies the district hires them. We have boarded the same bus to start this program* (*Connect* Focal Person, Kilombero CHMT, 2012).

In addition, the WAJA service delivery package included both preventive and basic curative care for children as a measure of "first aid" to handle simple illness. As a village supervisor from Rufiji explains, this combination helped address communities' perceived needs.

> *Before WAJA your child may be sick but [parents] are not aware because lack of education. Then it becomes emergency, so the child will get more sick while you get the money, but now when the child is sick you can see it right away, and WAJA gives drugs as first aid* (Village Supervisor, Mangwi, Rufiji, 2013).

Facility staff also appreciated the blended service delivery package, which they felt helped rationalize recourse to clinics for care. They were also pleased that the training and eligibility requirements were compatible with how the local health system worked.

> *This project has been received well by the district because WAJA have received nine-month training, which means they can be hired and because of that their contracts were approved. This shows that [WAJA] are employed and will work as normal staff do.* (District Medical Officer, Rufiji, 2014).

However, as implementation continued, weaknesses of local health system constrained districts' absorptive capacity. Rather than intervene to ensure optimal implementation conditions at facilities, *Connect* relied on local health systems to adapt independently to the changes that were instigated by WAJA deployment. This frustrated WAJAs, who recalled situations such as "I educated a pregnant woman before delivery to go [deliver at hospital] but when she reached [the hospital] she observed that there was no service, and she told her husband 'there is not any service. Its better I deliver at home'" (WAJA, Mgomba North, Rufiji, 2013). Health care workers at facilities were upset by these failures. When WAJA referred sick children, staff at facilities, often, could not help:

*A child might have malaria scorching hot, but you can fail to get medicine, or you can get medicine but fail to get syringe. You tell [the caregiver] to buy [the missing item] because we do not have enough working equipment* (Health Facility Supervisor, Mngeta, Kilombero, 2013).

The fragility of logistics systems illuminated a deeper divergence between individuals involved in the intervention. Whereas *Connect* staff and district counterparts deployed WAJA to empower communities and connect them to the health system, community members felt WAJA should act as 'doctors' that address needs unmet due to the weak health system. According to one WAJA: "The difficulty is that people see me as a doctor. When I refer the patient. . . They ask, 'why go to the facility where we get nothing if we chose you to work here for us?'" (WAJA, Lukolongo, Ifakara, 2012). Mothers voiced that perspective: "I would like WAJA to be given more trainings so that it will not be necessary to go to the hospital because when we go there, there are no medications" (Mother, Mgomba North, Rufiji, 2014).

Fig 5 illustrates the stakeholder causal pathway. Preconditions for the implementation effectiveness were not in place and the strategy did not contain elements that strengthened absorptive capacities of support system actors to adapt to the changes that ensued after WAJA deployment. Because of this, the targeted change mechanism, i.e., the 'intervention-values-systems' fit, was never activated. In turn, proximal and distal outcomes did not arise as intended. See Fig 5.

**WAJA level.**   In addition, *Connect* sought to build capacity, inspire motivation and, in doing so, help establish PHC implementation readiness in communities. These interventions focused on WAJA, primarily, and included training, employment and remuneration, and clinical and community supervision. Throughout the trial, these inputs had their intended effect: beyond the training and deployment of WAJA, being paid and accorded an 'organizational identity' that motivated WAJAs' productivity and commitment:

*The thing that pushed me is that I had no work [before becoming a WAJA] and I had received little education and I observed that the community was struggling. . . Now when we pass through the community [community members] recognize us as ones that can help them. . . This motivates me* (WAJA, Kisawasawa, Ifakara, 2012).

However, over time, three contextual factors affected WAJAs' sense of 'organizational citizenship'. Chronic failures in the supply chain in the later years of the project left WAJA feeling betrayed: "It is as if [the intervention] has entered the government system. First, they will replace [bicycle] tires, then they say they will replace medicines. But, when you wait you do not get the promises that they told us" (WAJA, Lumemo, 2014).

Relationships with health facility supervisors were crucial to WAJAs' motivation: "He always comes to ask us about the challenges we face and keeps regularly in touch to see if we need help with a patient. This is what motivates us, too" (WAJA, Minepa, Ifakara, 2013). Yet, there were frequent lapses in supervision, which supervisors attributed to excessive workload.

*We have been given transportation, which gives us motivation to do our work effectively with the WAJA, at the village, but we do not get enough support with our services. Who can provide our services if we are [in the community]?* (Health Facility Supervisor, Mlabani, Ifakara, 2012).

WAJAs' success also depended on their relationships with village leaders. WAJA benefited when village authorities helped mobilize households, enforce community health rules, and solve complex problems. One WAJA reports:

*Where I come from, it's the village health committee that makes decisions and implements the fines. So, as I pass through the community to inspect households, which are supposed to have a latrine toilet, I find some households have difficulty with this. When I go to this household, they might refuse or chase [me] away, so I report this so that the committee can help the household or charge fines.* (WAJA, Lumemo, Ifakara, 2013).

However, WAJA did not always experience productive collaboration with village leaders:

*In my village, the government does not cooperate. The public doesn't trust the executive. We had the lack of cupboards for storing the medicine. This was troubling [me] until there came some doctors who strongly rebuked [the Village Executive Officer]. Another thing the meetings. . .. The meetings should be held every three months, but since last year we have not had one.* (WAJA, Mangwi, Rufiji, 2013).

Ubiquitously, village stakeholders complained that while WAJA were remunerated, Village Supervisors and other members of village governments were not. As one Village Supervisor lamented: "We fail to do our job because we don't have allowance, and the WAJA will not listen to you, the supervisor, as you are only a volunteer while that one is getting paid" (Village Supervisor, Nyambunda, Rufiji, 2012).

In addition, geographic and logistical barriers impeded WAJA performance. The wetland terrain of the Rufiji delta was particularly vulnerable to extreme rainfall: "What disrupts our schedule is the weather condition. There are some hamlets which you cannot reach because of rivers that get created by monsoon rains" (WAJA, Nyambunda, Rufiji, 2012). WAJA deployed to expansive, rural communities struggled to meet coverage requirements: "What hinders us is the distances from some homes and the rest of the community. Sometimes bicycle might be damaged, and the distance is so far, in the bush" (WAJA, Lukolongo, Ifakara, 2012).

Pursuit of the WAJA-level causal pathway was achieved with mixed success. Although the preconditions for implementation effectiveness were in place for some of the program, lapses occurred that were addressed by the *Connect Project* when districts could not. WAJA often struggled to overcome difficult community dynamics and environmental constraints. Nevertheless, triangulation of qualitative and quantitative findings suggests that, despite challenges, WAJA maintained their confidence and motivation, and implementation readiness in communities, which helped achieve some of the intended distal MNCH outcomes. See Fig 6.

**Organizational level.** Finally, at the organizational level, *Connect* sought to strengthen systems to adopt and manage the WAJA intervention. This started with the IHI providing financial and embedded technical assistance to CHMT during the first two years of the trial. Although *Connect* periodically backstopped districts in the provision of essential working materials to WAJA from 2013–15, the strategy largely failed at incorporating the intervention in the government system. One stakeholder commented: "The challenge is how we can incorporate [WAJA] in the health system. That challenge is too big for us, as it requires cooperation between us, the Ministry of Health as well Ministry of Regional Government, which now we do not see" (District Medical Officer, Ulanga, 2014).

In addition, district leaders felt that the CHW sub-system was simply too costly and complicated to adopt in two years.

*WAJA program has come here as pilot and will not last forever, and we need more resources. Our budget from the basket fund should be spent on services at facilities. 'OS' (Other Sources), which we received from the central government, this can go to [the WAJA intervention] . . .*

*But if you spend on office affairs, fuel, the allowances, salaries, and everything in the adminis-tration activities, it is not enough.* (District Medical Officer, Rufiji, 2014).

It follows that, during 2013–2015, WAJA depended on support from the IHI to sustain implementation.

*As a district I can't say that we can accommodate WAJA so that they can be sustainable. The WAJA are still doing their duties as usual and I still visit them to deliver supplies and do fol-low up, but for most things now we wait for it to get done [at the IHI] with Connect* (*Connect* Focal Person, Kilombero, 2014).

As these stakeholders discussed, the combination of weak health systems and an inadequate district support strategy contributed to breakdown in the organizational level causal pathway. Preconditions for strategic success were never in place and *Connect's* financial and technical support to districts was insufficient vis-à-vis the costs and complexity of integrating the WAJA into local health systems. See Fig 7.

## Discussion

This study evaluated the effect of WAJA deployment on MNCH service utilization, childhood morbidity and access to sick childcare, and explained how these findings were generated. The quantitative analysis demonstrates that the intervention failed to connect communities to the formal health system. In fact, there is some indication that the intervention might have inhib-ited utilization of ANC and PNC at clinics, possibly since WAJA performed the information, education and counseling components of these services in households, though these findings were not statistically significant. However, WAJA deployment was associated with childhood morbidity reduction as well as increases in timely access to appropriate of sick childcare. The qualitative analysis elucidates how the implementation strategy did and did not affect intended outcomes and the determinants of that process. Altogether, this study provides an opportunity to hypothesize about the relationship between CHW interventions, the mechanisms through which they change proximate outcomes, how contextual conditions shape these processes, and how this affects MNCH.

### Intervention source, adaptability, and visibility of results

By engaging communities in cocreation of the intervention, adapting training procedures, and using community members to provide MNCH care, *Connect* elicited the perception that the WAJA intervention had been internally developed and that lifesaving interventions had been adapted and configured in the social environment to meet communities' needs. Early-stage resistance subsided as skeptics became familiar with the WAJA and observed their impact. These findings add to literature on acceptability and diffusion of innovation, which empha-sizes similar factors [71, 72]. This analysis suggests that these mechanisms helped *Connect* achieve critical proximal outcomes, such as legitimacy, household members satisfaction, and community acceptance, which, we conjecture, facilitated caregivers' uptake of health behaviors that, in turn, led to positive health effects, importantly the reduced risk of childhood illness. Additional more focused research is needed to better understand the mechanisms through which community-based programming triggers acceptability, and to demonstrate whether this is associated with behavior change.

However, our analysis found that the process used to determine the WAJA service delivery package did not sufficiently engage communities and include some components that were

valuable to them, and was, thus, a notable barrier to acceptability. Future research ought to investigate the feasibility of strategies that integrate citizen accountability structures into PHC policy and implementation processes [73, 74]. Furthermore, the qualitative analysis found that acceptability was greater in communities with strong leadership that proactively supported WAJA [34, 75, 76]. Future research should focus on ways to strengthen these structures as a step toward incorporating communities into wider health systems.

### Intervention-values-systems fit

The qualitative findings bring into focus how weaknesses of the health system stymied creation of such an 'intervention-values-systems' fit [70, 77]. Whereas *Connect* managers viewed WAJAs as '*connecters*' deployed to link communities to the health system and motivate them to take prevention and promotion of health into their own hands, community members saw WAJA as ambulatory doctors in place to compensate for a health system that failed them. This divergence of perspectives and misalignment of the intervention design, communities' perceived needs, the readiness of the health system to routinely provide MNCH care, and its limited absorptive capacity produced a climate of implementation replete with referral noncompliance, ill-prepared WAJA and health care workers, and frustrated communities. This finding has implications for CHW programs that are situated in weak health systems. To succeed in these environments, CHW implementation strategies should incorporate systems strengthening components that address root causes of communities' suboptimal use of facility-based care. We hypothesize that this will help establish an implementation climate of greater compatibility between the roles of CHW, communities' perceptions and willingness to use the CHW intervention, and the capacity of delivery systems to meet expectations. In doing so, future CHW programs may succeed in areas where *Connect* did not, for example strengthening linkages between communities and the health system and increasing use of facility-based MNCH services.

Previous research has focused on compatibility between intervention features and capacities of delivery systems to learn and adapt [78, 79], existing workflows and systems in the adopting organization [80, 81], and implementer characteristics [82, 83]. In addition, studies have reported on the adaptability of interventions vis-a-vis value systems conditioned by religion, traditional social and communication networks and diverse cultures [84–87]. This analysis illuminates how characteristics of systems condition the ways in which stakeholders perceive, value, and use interventions; and how this affects the introduction of CHW. Future research should explore these relationships, and how they shape the prospects of introducing evidence-based interventions in health systems.

### Readiness for implementation

*Connect's* pragmatic objective to build capacity of the system to adopt and implement the WAJA intervention largely failed, a finding that is consistent with examples from many CHW initiatives which encountered organizational and sustainability challenges [88]. In particular, CHW programs that emphasize IMCI have struggled to maintain capabilities to implement essential components of integrated care systems, such as supervision, remuneration or incentivization of cadres and supply chain logistics [89–92]. Future research should focus on how to strengthen public sector health system leadership and coordination capacities as a precursor to extending the reach of PHC programs to the community-level [58, 93].

### Motivation and organizational commitment

Beyond building WAJA knowledge and skills, *Connect* succeeded at engendering the commitment of the cadre by giving them contracts and remuneration for performance, extending

formal supervision processes to the community, and tapping into existing community structures to avail WAJA with necessary support. The qualitative analysis illustrated how this triggered motivation, confidence, and enhanced organizational citizenship behaviors. These mechanisms not only made WAJAs' work more feasible and enabled their reach in communities. Moreover, they withstood adverse contextual influences, such as lapses in health system functionality, inconsistent support from community leaders and geographic and weather-related barriers and enabled the achievement of positive health effects. Based on this, we conjecture that programs that blend efforts to professionalize CHW, facilitate opportunities for supervision and support, and ensure the functionality of logistics systems elicit levels of hard work, motivation, and confidence from CHW that, in turn, leads to desirous performance outcomes.

In LMICs, there is dearth of research on the dynamic interplay between individuals and their organizations and how this affects implementation. Our finding that the professionalization of WAJA enhanced their willingness to work hard echoes earlier studies which highlight the relevance of personal growth, professional development, and both working and social relationships to CHW motivation [94]. It is noteworthy that WAJA motivation was inhibited when incentivization schemes came into tension with social relationships and hierarchies. Bhattacharyya et al. make a distinction between factors that motivate CHW and factors that motivate others to support and sustain CHW [95]. Indeed, 'complementary incentives' are an important consideration when establishing incentivization schemes not just for CHW but for sustaining the wider 'community health system' [41].

## Integration of communities into the health system

The qualitative analysis illuminated contextual factors that impeded organizational adoption of the intervention, an objective of *Connect*. Notably, the costs and complexity of doing so was high relative to extant capacities of systems to learn and adapt. Earlier studies have underscored the need to anticipate these demands and address them proactively to ensure that delivery systems are poised to depart from existing practices, mindful of the intricacy and number of steps required to do so and a realistic sense of the increase in organizational target units that must be reached by implementation [71, 96–98]. Previous research has also demonstrated the importance of external policies, incentives, or regulations, led by governments or other central entities, that instruct, motivate and channel direct and indirect support to implementers for the uptake and spread of interventions in health systems [37, 43].

## The study suffered from some limitations

Since it was not possible to blind participants to their treatment status, it is possible that those who were residents of comparison communities might have visited nearby intervention villages to receive care, or care for their children, from WAJA, which may have biased results. In addition, recall bias may have interfered with measurement in the surveys, which relied on participants' recollection of care received, in some instances, as far back as two-years prior to data collection. The analysis conducted multiple statistical tests and, therefore, faces risks associated with multiplicity of analyses and outcomes. Qualitative findings might have been affected by 'social desirability' bias in that respondents might have adulterated their responses to please the team that led the intervention. Child survival in the study districts during the time of and surrounding *Connect* had been improving at a pace more rapid than was average for much of Tanzania. Thus, it is not clear if the same results would have been posted had WAJA been deployed in other districts. Finally, our discussion features hypotheses of the linkages between intervention components, mechanisms of change, implementation

determinants, proximal outcomes, and health effects. Although these propositions are substantiated by our analysis, additional research is needed to understand these relationships more deeply and test them.

## Conclusion

The evaluation of *Connect*, a pragmatic trial with embedded implementation research, shows the outcomes and processes of introducing CHW in the realities of a health system struggling to maintain effective coverage of facility based PHC. The attribution of null effects to systemic weaknesses points to the need for strategies that strengthen and align community and health systems' core capacities, as well as their abilities to learn, adapt and integrate best evidence-based interventions. In the case of WAJA in Tanzania, there is evidence that suggests that, by addressing this gap, it is possible to accelerate child mortality reduction and improve MNCH. If policymakers, implementation teams and communities, and researchers work together with a common vision of community health systems strengthening, they can help achieve universal health care in Tanzania and similar settings.

## Supporting information

**S1 Checklist. CONSORT checklist.**
(DOCX)

**S1 Table. Socio-demographic balance of baseline sample (2011).**
(DOCX)

## Acknowledgments

We recognize the contributions of Ruth Wilson, Awena Malendo, Mustafa Njozi and Ahmed Hingora. Finally, we recognize with gratitude the leadership and importance of staff and supporters from Kilombero, Rufiji and Ulanga districts, including staff of their respective Council Health Management Teams and authorities of the villages in which this research was conducted.

## Author Contributions

**Conceptualization:** Colin Baynes, Kassimu Tani, Bryan J. Weiner.

**Data curation:** Colin Baynes, Amon Exavery, Kassimu Tani, Gloria Sikustahili, Hildegalda Mushi, Jitihada Baraka, Kate Ramsey.

**Formal analysis:** Colin Baynes.

**Investigation:** Amon Exavery, Kassimu Tani, Gloria Sikustahili, Hildegalda Mushi, Jitihada Baraka, Kate Ramsey.

**Resources:** James F. Phillips.

**Supervision:** Almamy Malick Kanté.

**Validation:** Kenneth Sherr, Bryan J. Weiner, James F. Phillips.

**Writing – original draft:** Colin Baynes.

**Writing – review & editing:** Colin Baynes, Almamy Malick Kanté, Amon Exavery, Kassimu Tani, Gloria Sikustahili, Hildegalda Mushi, Jitihada Baraka, Kate Ramsey, Kenneth Sherr, Bryan J. Weiner, James F. Phillips.

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
