## [Decision Letter · Decision Letter 0]

20 Jul 2023

PGPH-D-23-00960

The implementation and effectiveness of multi-tasked, paid community health workers on maternal and child health: A cluster-randomized pragmatic trial and qualitative process evaluation in Tanzania.

Dear Dr. Baynes,

Thank you for submitting your manuscript to PLOS Global Public Health. After careful consideration, we feel that it has merit but does not fully meet PLOS Global Public Health’s publication criteria as it currently stands. Therefore, we invite you to submit a revised version of the manuscript that addresses the points raised during the review process.

You can ignore the first comment that reviewer 1 provided.

We look forward to receiving your revised manuscript.

Kind regards,

Ferdinand Mukumbang, PhD

Academic Editor

Journal Requirements:

1. Please include an explanation for the retrospective CT registration and confirmation that all related CTs are registered.

2. In the online submission form you indicate that your data is not available for proprietary reasons and have provided a contact point for accessing this data. Please note that your current contact point is a co-author on this manuscript. According to our Data Policy, the contact point must not be an author on the manuscript and must be a third party. Please revise your data statement to a non-author institutional point of contact, such as a data access or ethics committee, and send this to us via return email. Please also include contact information for the third party organization, and please include the full citation of where the data can be found.

3. Please provide a/amend your detailed Financial Disclosure statement. This is published with the article. It must therefore be completed in full sentences and contain the exact wording you wish to be published.

State the initials, alongside each funding source, of each author to receive each grant.

4. We have noticed that you have uploaded Supporting Information files, but you have not included a list of legends. Please add a full list of legends for your Supporting Information files after the references list. 

5. Some material included in your submission may be copyrighted. According to PLOS’s copyright policy, authors who use figures or other material (e.g., graphics, clipart, maps) from another author or copyright holder must demonstrate or obtain permission to publish this material under the Creative Commons Attribution 4.0 International (CC BY 4.0) License used by PLOS journals. Please closely review the details of PLOS’s copyright requirements here: PLOS Licenses and Copyright. If you need to request permissions from a copyright holder, you may use PLOS's Copyright Content Permission form.

Potential Copyright Issues:

Figure 1: please (a) provide a direct link to the base layer of the map (i.e., the country or region border shape) and ensure this is also included in the figure legend; and (b) provide a link to the terms of use / license information for the base layer image or shapefile. We cannot publish proprietary or copyrighted maps (e.g. Google Maps, Mapquest) and the terms of use for your map base layer must be compatible with our CC-BY 4.0 license. 

Additional Editor Comments (if provided):

Reviewers' comments:

Reviewer's Responses to Questions

**Comments to the Author**

1. Does this manuscript meet PLOS Global Public Health’s publication criteria? Is the manuscript technically sound, and do the data support the conclusions? The manuscript must describe methodologically and ethically rigorous research with conclusions that are appropriately drawn based on the data presented.

Reviewer #1: Yes

Reviewer #2: Yes

2. Has the statistical analysis been performed appropriately and rigorously?

Reviewer #1: Yes

Reviewer #2: I don't know

3. Have the authors made all data underlying the findings in their manuscript fully available (please refer to the Data Availability Statement at the start of the manuscript PDF file)?

Reviewer #1: No

Reviewer #2: Yes

4. Is the manuscript presented in an intelligible fashion and written in standard English?

Reviewer #1: Yes

Reviewer #2: Yes

5. Review Comments to the Author

Reviewer #1: Interesting topic with a strong methodological approach.

Comments

1- P6: Figure 1 , P11: Figure 2 P 23, p 26, p 29, all those figures are announced in the text but not shown. A decision need to be made to insert the figures in the text or to leave them as an annex

2- P 14 Method: This sentence may be confusing : “We purposively sampled four nodes in Ifakara and two in Rufiji. We sampled two urban nodes in

1- Ifakara and Rufiji, respectively, and four rural nodes.” Is it 1 urban node in Ifakara and 1 in Rufiji? Make this part clearer. Additionally, when presenting the study setting, it will help to announce that some of the sites are urban , others rural and identify in what districts they are.

2- P16: It could have been interesting to know how the result of the quan and qual linkage will be presented.

3- P17: “There is some indication that the intervention might have inhibited utilization of ANC 375 and PNC at clinics, possibly since WAJA performed the information, education, and counseling 376 components of these services in households, though all findings are statistically insignificant.” This segment highlighted in red will fit better in the discussion part than the results section.

4- P 31 in the sentence “and was, thus, was a notable barrier to acceptability”. Delete one of the “was”

5- Considering the study limits, does the fact that the study participants was not blinded about their treatments can influence the evaluation results?

Reviewer #2: This is an interesting paper and I applaud authors’ work.

I have few comments below

Authors report that the ‘community health worker program reduced incidence of childhood illness and improved access to timely and appropriate sick childcare; however, there was no effect on maternal, newborn and child health service utilization’—this statement needs be clearly explained as this can add confusion to readers. How is childhood illness incidence reduced when there was no effect on service utilization?

And then at line 55: Authors statement reads ‘The lack of……among community health workers’. You have attributed it to lapses in the availability of needed care at facilities and or working materials among community health workers. So, it could be a major limitation compromising the outcomes including reduced childhood illnesses? In other words, Why did this factor not affect the childhood illnesses?

Since the CHW deployment also has heterogeneity in terms of who they are and how they could affect the outcome, did you account the individual differences, their aptitude, performance, and catchment area impacts?

It looks like the CHW training and interventions experienced a lot of changes in terms of supervision, did that affect the outcome? Has that been accounted, particularly on how that may have had impact on their performance and ultimately outcomes?

Also can you explain why IHI withdrew from supporting implementation?

6. PLOS authors have the option to publish the peer review history of their article (what does this mean?). If published, this will include your full peer review and any attached files.

**Do you want your identity to be public for this peer review?** For information about this choice, including consent withdrawal, please see our Privacy Policy.

Reviewer #1: No

Reviewer #2: No

---

## [Editor Report · Decision Letter 1]

22 Aug 2023

The implementation and effectiveness of multi-tasked, paid community health workers on maternal and child health: A cluster-randomized pragmatic trial and qualitative process evaluation in Tanzania.

PGPH-D-23-00960R1

Dear Dr Colin Baynes,

We are pleased to inform you that your manuscript 'The implementation and effectiveness of multi-tasked, paid community health workers on maternal and child health: A cluster-randomized pragmatic trial and qualitative process evaluation in Tanzania.' has been provisionally accepted for publication in PLOS Global Public Health.

Best regards,

Ferdinand Mukumbang, PhD

Academic Editor